# How Psychophysical Stress Can Mediate the Effects of Anxiety and Depression on the Overall Quality of Life and Well-Being in Women Undergoing Hereditary Breast Cancer Screening

**DOI:** 10.3390/cancers16213613

**Published:** 2024-10-26

**Authors:** Anita Caruso, Caterina Condello, Gabriella Maggi, Cristina Vigna, Giovanna D’Antonio, Laura Gallo, Lara Guariglia, Antonella Savarese, Giulia Casu, Paola Gremigni

**Affiliations:** 1Psychology Unit, IRCCS “Regina Elena” National Cancer Institute, 00144 Rome, Italy; anita.caruso@ifo.it (A.C.); gabriella.maggi@ifo.it (G.M.); cristina.vigna@gmail.com (C.V.); giovanna.dantonio@ifo.it (G.D.); laura_gallo@hotmail.it (L.G.); 2Department of Oncohematology, Federico II University of Naples, 80138 Naples, Italy; caterina.condello@unina.it; 3Department of Oncology, IRCCS “Regina Elena” National Cancer Institute, 00144 Rome, Italy; antonella.savarese@ifo.it; 4Department of Psychology, University of Bologna, 40127 Bologna, Italy; giulia.casu3@unibo.it (G.C.); paola.gremigni2@unibo.it (P.G.)

**Keywords:** breast cancer genetic counseling, BRCA1/2 mutations, emotional distress, psychophysical stress, quality of life, well-being, mediation

## Abstract

Women at risk of carrying a genetic mutation report higher levels of distress compared to the general population. The aim of this study was to evaluate the mediating role of psychophysical stress in the relationship of anxiety and depression with quality of life and well-being in women undergoing genetic counseling for BRCA1/2 mutations. The results indicate that psychophysical stress, generated by emotional distress, significantly influences the perceived mental well-being and overall psychophysical health. This paper emphasizes the need for integrated care, addressing psychological distress alongside physical health and recognizing the multifaceted impact of hereditary breast cancer screening on patients’ overall quality of life and well-being.

## 1. Introduction

Advances in molecular biology have identified certain hereditary factors within the population of women with breast cancer that predispose them to this oncological condition. It is estimated that 18% of breast cancer cases occur due to hereditary factors while 13% are caused by the BRCA1/2 genetic mutation [1]. Women who are potentially at risk of carrying this genetic mutation may be referred to a genetic counseling pathway during screening phases or oncological treatments. This pathway includes both pre- and post-test consultations. In the pre-test phase, the motivations for seeking genetic counseling are explored, particularly to uncover expectations and/or preconceptions. Additionally, a reconstruction of the individual’s personal and family medical history is conducted [1]. This model aims to achieve informed consent for genetic testing. Following the genetic test, individuals who carry the BRCA1/2 pathogenic variant are offered primary and/or secondary prevention measures to reduce their hereditary cancer risk. These measures include both intensive clinical–instrumental surveillance as well as prophylactic surgery.

Throughout the genetic counseling process, attention is given not only to medical data but also to the psychosocial characteristics and issues of the patients. This approach aims to protect the psychological well-being of these individuals and assess how these factors influence patient management. It is known that women at high risk of genetic mutation often carry a significant emotional burden due to their family cancer experiences. They may be overwhelmed by the information received during the initial phases of genetic counseling. The perception of being at risk can affect their quality of life and significantly impact their emotional states [2]. Conditions of distress, such as excessive anxiety or worry, can reduce the effectiveness of counseling interventions [3,4]. This is because high levels of emotional distress can make it difficult for patients to process information and make informed decisions.

Current studies investigating the psychosocial characteristics of women potentially carrying a genetic mutation have revealed several key findings. Women at risk of carrying a genetic mutation report higher levels of distress compared to the general population. This distress is primarily expressed through anxiety symptoms while their depression symptoms are comparable to those of the general population [5,6,7,8]. Within this population, certain groups are more vulnerable than others. Patients with a past or current history of cancer exhibit greater psychological distress compared to healthy people [2,9]. For long-term cancer survivors, distress persists throughout the entire course of the disease [10]. Studies have highlighted a relationship between symptoms of depression and cognitive deficits, particularly in logical, verbal, and visual domains, as well as in short-term memory tasks [11]. The relationship between distress levels and education is unclear: Dorval et al. [7] found that more educated women experience less distress whereas Oliveira et al. [5] reported that women with higher education levels experience greater distress and a poorer quality of life. Distress levels vary according to age. Women under 40 years of age experience higher degrees of distress compared to older women [12,13]. It is likely that the impact of a genetic mutation diagnosis is more severe at a younger age when the risk of developing cancer is considerably lower [14].

The literature confirms that the emotional distress experienced by patients undergoing genetic counseling affects their quality of life across all domains: physical, psychological, relational, environmental, and general [2,5]. Within the psycho-oncological perspective, quality of life encompasses not only physical health, symptoms of illness, or side effects of therapy but also psychological and socioeconomic factors. Key components include physical health, which involves the absence of disease, physical fitness, and the ability to perform daily activities; mental health, which encompasses emotional, cognitive, and social well-being; economic stability; and environmental factors such as living conditions, access to healthcare, education, and recreational activities. Quality of life is a crucial issue at all stages of the cancer disease trajectory. For patients with breast cancer, a poor quality of life is evident both during the treatment period and in the years following it [15,16]. Many factors can influence quality of life. For example, a family history of cancer is linked to a lower quality of life in the relational domain, and younger age is associated with poorer quality of life [5]. This study focused particularly on the relationship between quality of life and psychological distress. The literature indicates that in breast cancer, quality of life is influenced by anxiety and depression [15,16]. Conversely, lower levels of depression are associated with a better perceived quality of life [17]. Even in the later stages of genetic counseling, high levels of anxiety and depression continue to reduce the health-related quality of life [18].

Studies assessing the psychological distress of women potentially carrying a genetic mutation through a long-term prospective design have revealed findings of considerable relevance for both clinical practice and research. High levels of distress and a poor quality of life in the pre-test phase are strong indicators of clinically significant distress following the diagnosis of a genetic mutation. Pre-test distress is the greatest risk factor for clinically significant conditions both one year after the genetic mutation diagnosis and in the long term [19,20]. High levels of distress, complicated grief conditions, the number of relatives affected by cancer, and illness representation are the factors that best explain distress in this population both in the pre-test phase and six months after the results, regardless of the test outcome [21,22]. These findings highlight how distress can be an important predictor for the psychological health of this population both in the initial phases of genetic counseling and in the phases following the diagnosis of a genetic mutation. It is known that psychological distress can become chronic, destabilize family relationships, lead to a poor quality of life [19,23,24], and influence prevention and treatment choices. However, psychological distress in the pre-test phase is not always adequately assessed by physicians [25] and has been poorly investigated in the literature, where studies have predominantly focused on distress resulting from genetic test outcomes.

While quality of life is a broad concept that encompasses various aspects of an individual’s overall well-being and essentially reflects how well an individual can enjoy and participate in life, psychophysical well-being has a more specific focus. It refers to the harmonious balance between an individual’s psychological—mental and emotional—and physical health, emphasizing the interplay between the mind and body. A study on chronic dialysis patients has reported small to moderate correlations between psychophysical well-being and components of quality of life, indicating that the two constructs are slightly overlapping but remain independent from each other [26]. Therefore, both quality of life and psychophysical well-being are important for a holistic understanding of an individual’s health and happiness, which is expected to be adopted within psycho-oncology [27]. However, few studies have focused on psychophysical well-being as an outcome in the context of cancer. An Italian study has highlighted that patients with chronic cancer pain have a lower level of psychophysical well-being than patients with other chronic pains [28].

Another important construct that has been extensively studied in relation to cancer is the perceived experience of stress [29]. This construct refers to an individual’s feelings about the overall stressfulness of their life. In women with breast cancer, perceived stress negatively impacts psychological well-being and quality of life [30]. Research into stress factors in women who may carry a genetic mutation reveals that both familial and personal cancer diagnoses significant contribute to stress [31]. Having a close relative with cancer can be as stressful as having a personal history of cancer [32]. However, the role of perceived stress in relation to the psychophysical condition in individuals attending cancer genetic testing has not yet been investigated.

Building on the above-mentioned constructs, the aim of this study was to deepen the understanding, through mediation models, of the relationship between psychological distress, stressful life experiences, and quality of life and well-being in women in the initial phase of genetic counseling. Studies using mediation models as a method of investigation have been scarce and have mainly focused on other variables. For example, in patients with psoriasis, alexithymia reduces the perception of mental quality of life through the mediation of psychological distress in terms of anxiety and depression [33]. In patients with cancer, health-related stress diminishes quality of life through the mediation of perceived stress [34]. In breast cancer patients, higher symptoms levels are associated with poorer psychophysical well-being mediated by subjective stress appraisals [30]. In the context of genetic testing for hereditary breast cancer, higher levels of anxiety and depression increase cancer worries and risk perception through the mediation of health fears [35].

The uniqueness of this study lay in investigating, for the first time, the relationship between distress, perceived stress, quality of life, and physical and mental well-being in patients undergoing genetic screening for breast cancer during the initial stages of genetic counseling, before the test results were available. Understanding the psychophysical conditions of patients awaiting genetic testing results could aid in the subsequent phase of communicating the results and providing recommendations for managing any positive outcomes.

### Objective

In our study, we explored how anxiety, depression, and stressful life experiences influence both physical and mental quality of life as well as psychophysical well-being. We focused on three mediation models where psychophysical stress acted as a mediator between trait anxiety and depression mood as predictors and mental quality of life, physical quality of life, and psychophysical well-being as outcome variables. To better understand these relationships, we considered sociodemographic and clinical characteristics as potential confounding factors.

## 2. Methods

### 2.1. Procedure

The study sample was recruited from the genetic counseling clinics of the Regina Elena National Cancer Institute in Rome and the University Federico II in Naples, Italy. Eligible participants were women aged 18 years or older, awaiting genetic counseling for the BRCA1/2 mutation, with at least one first-degree relative with breast and/or ovarian cancer. Excluded from the study were women under 18 years of age, women from families already tested for a BRCA1/2 diagnosis or who had previously undergone genetic counseling, and women awaiting cancer screening results. Eligible women signed an informed consent form, and the research project was approved by the ethics committee (Lazio District 5 Territorial Ethics Committee—Verbal Extract n. 10 of 20 December 2023—Trial Register Experiments N. 76/IRE/23) of the Regina Elena National Cancer Institute. The total number of participants was 193, recruited from the two hospitals with the following proportions: 89% (n = 171) from Regina Elena National Cancer Institute in Rome and 11% (n = 22) from Federico II University in Naples (binomial test *p* < 0.001).

### 2.2. Measures

Participants underwent an assessment that included sociodemographic and clinical details along with self-report questionnaires.

Sociodemographic data included age at test completion, education level (primary and middle school, high school, and university/tertiary), marital status (married/cohabiting, yes–no), parental status (having children, yes–no), and occupational status (having a job, yes–no). Clinical information encompassed a breast cancer diagnosis (yes–no), familiarity with breast cancer (yes–no), and the number of relatives affected by breast cancer.

Furthermore, the battery of assessment included the following questionnaires.

The Short-Form 12 Health Survey (SF-12) [36] assesses both physical and mental quality of life. It consists of 12 items organized into two subscales: the Physical Component Scale (PCS) and the Mental Component Scale (MCS). These subscales measure the following dimensions: limitations in physical, social, and usual role activities because of physical or emotional health problems; bodily pain; general mental health; vitality (energy and fatigue); and general health perceptions. Examples of items are “Does your health currently limit you from carrying out activities of moderate physical effort, such as moving a table, using the vacuum cleaner, playing bowls, going for a bike ride?” (Physical Component Scale) and “In the last 4 weeks, due to of your emotional state, did you perform less than you would have liked?” (Mental Component Scale). These subscales are expected to be weekly intercorrelated. The questionnaire uses various rating scales, including 6-point, 5-point, and 3-point scales, as well as categorical yes–no answers. Scores are derived from weighted combinations of the same items, meaning each item contributes to the total score based on its importance. The total scores for each scale range from 0 to 100, with lower scores indicating a poorer quality of life. The study utilized the validated Italian version of the SF-12 [37] to measure health-related quality of life.

The State–Trait Anxiety Inventory (STAI) [38] consists of two subscales, each comprising 20 items. The first subscale (STAI-X1) assesses state anxiety, which reflects the anxiety a person experiences at the moment of questionnaire completion. The second subscale (STAI-X2) measures trait anxiety, representing a more enduring anxiety state that individuals tend to experience in most daily situations. Examples of items are “I feel pleasant” and “I feel indecisive”. The response scale for this subscale ranges from “rarely” (1) to “almost always” (4). For the purposes of the present study, we employed the STAI-X2 scale in the Italian translation and adaptation [39].

The Cognitive Behavior Assessment—Hospital form (CBA-H) [40] has been developed for a rapid psychological assessment in the context of health and somatic diseases. It includes 4 cards that cover subjective, emotional, and behavioral problems associated with a suspected or diagnosed somatic disease. In this study, we used Card B, which comprises 23 items measuring three different factors that refer to psychophysical sensations, emotions, and perceptions over the preceding three months: depression mood (DM), with 10 items; psychophysical well-being (PW), with 6 items; and psychophysical stress (PS), with 7 items. We selected this questionnaire because it provides concise measures of the main variables of interest in this study. Specifically, the DM subscale refers to a decrease in mood and performance in a depressive sense. Examples of items include “My interest in things I enjoy has decreased”. We used this subscale along with the STAI to assess distress levels, focusing on symptoms of anxiety and depression. The PW subscale refers to a perceived state of well-being at both psychological and physical levels. Examples of items are “I have slept well” and “I felt relaxed and serene”. We used this subscale together with the SF-12 to provide a more holistic representation of the study outcome. The PS subscale refers to the perception of having experienced a stressful and exhausting life over the last three months. Examples of items include “I got tired easily” and “The last period has been strongly stressful”. We used this subscale to address the mediator in our models. Responses are scored as true/false, with a score of 1 or 0 assigned. Higher scores indicate negative conditions, except for PW, where higher scores represent better well-being. The CBA-H battery has been developed in Italy and is widely used in the field of health psychology, including in oncology. It has been particularly valuable in hospital settings, allowing differentiation of emotional states and behavioral changes related to recent disease recognition or hospitalization. The CBA-H has been employed with patients experiencing acute or chronic organic diseases during the early days of hospitalization, as well as with outpatient individuals facing significant health events, such as receiving a diagnosis, and those participating in primary prevention programs. CBA-H has been recently used with women attending genetic counseling for hereditary breast cancer [26].

### 2.3. Data Analysis

Descriptive statistics were employed to summarize the sociodemographic and clinical characteristics of the participants and the scores from the questionnaires used. Reliability was assessed for each measure using Cronbach’s alpha or Kuder Richardson’s formula for dichotomous items, with values greater than 0.69 considered acceptable and values exceeding 0.80 considered good.

Preliminary Pearson’s correlation analyses were conducted for all psychological variables to verify the assumptions for running the mediation models. Specifically, we expected significant intercorrelations between the predictors, the mediator, and the outcomes. In the mediation models, we also considered sociodemographic and clinical aspects as potential confounding variables. To select the confounding variables to be included in the mediation models, we preliminarily analyzed Pearson’s correlations between these variables and the outcomes, considering only those that were statistically significant. Subsequently, we performed a multiple linear regression analysis to reduce the number of confounding variables by selecting those that had had significant effects on the outcomes.

Three mediation models were examined, with psychophysical stress as a mediator between predictors (trait anxiety and depression mood) and outcome variables (mental quality of life, physical quality of life, and psychophysical well-being). Robust standard errors, robust confidence intervals, ML estimator, and Bonferroni correction for multiple analyses were used. The significance level was set at *p* < 0.05. All statistical analyses were conducted using JASP (version 0.18.3.0) [41].

## 3. Results

### 3.1. Participants

The 193 participants in the study were ranging in ages from 18 to 81 years. Most of them were highly educated, unmarried or not cohabiting with a partner, childless, and not employed. Among them, 49.2% had been diagnosed with breast cancer, 8.8% had a family history of breast cancer, and the number of relatives affected by breast cancer varied from none to eight (refer to Table 1).

### 3.2. Descriptive Statistics of Psychological Measures and Reliability

The descriptive statistics of the psychological variables are reported in Table 2. The reliability of the measures was acceptable/good, with Cronbach’s alpha exceeding 0.70 for all of them.

### 3.3. Preliminary Associations

Preliminary intercorrelations between the psychological variables were all statistically significant with moderate to high strength (as detailed in Table 3). Therefore, the assumption for including these variables in the subsequent mediation models was met.

Regarding the selection of potential sociodemographic and clinical confounding variables to include in the median models, a few preliminary correlations with the outcomes were statistically significant, with small to moderate strength (refer to Table 4). Specifically, age was negatively correlated with physical quality of life, education was positively correlated with mental quality of life and negatively with physical quality of life, having a diagnosis of breast cancer was negatively correlated with physical quality of life, and both breast cancer familiarity and the number of relatives with cancer were positively correlated with physical quality of life. The other sociodemographic variables were not correlated with the outcomes.

We conducted a multiple linear regression analysis to reduce the number of potentially confounding variables among those that were significantly correlated with physical quality of life. The model explained 17% of the outcome variance (adjusted R^2^ = 0.17; F_5, 185_ = 80.94; *p* < 0.001). Table 5 revealed that education and having a breast cancer diagnosis were the most influential factors, with a small positive effect and a moderate negative effect, respectively. The other variables had negligible effects on the outcomes. Therefore, education and having a diagnosis of cancer were included in the subsequent mediation models as confounding variables.

### 3.4. Mediation Models

In the first mediation model, we examined whether psychophysical stress acts as a mediator in the relationships between trait anxiety and depression mood (predictors) and mental quality of life (outcome) while also considering the education level as a confounding variable.

As observed from Table 6, psychophysical stress mediates both the relationship between trait anxiety and mental quality of life and the relationship between depression mood and mental quality of life. Specifically, higher levels of trait anxiety and depression mood contribute to increased psychophysical stress, which, in turn, reduces the perceived mental quality of life. However, while the mediation effect is partial in the case of the relationship between trait anxiety and the outcome, for depression mood, the mediation is full—meaning that introducing the mediator renders the direct effect of the predictor on the outcome no longer significant. The model explains 56% of the variance in mental quality of life (R² = 0.56) while the explained variance in psychophysical stress is 50% (R² = 0.50). The path coefficients can be observed in Figure 1. The two predictors are strongly associated with each other. Based on the data, it appears that education is a risk factor that weakly increases psychophysical stress and decreases mental quality of life.

In the second mediation model, our investigation focused on the potential role of psychophysical stress as a mediator between trait anxiety and depression mood, which were predictors, and physical quality of life, the outcome variable. This analysis also took into account the education level and having a breast cancer diagnosis as confounding variables. According to the results presented in Table 7, psychophysical stress does not serve as a mediator in this context. Notably, depression mood has a significant direct impact on physical quality of life whereas trait anxiety does not. The path coefficients, detailed in Appendix A Table A1, indicate that the education level slightly increases psychophysical stress, as previously observed in the first mediation model, and enhances physical quality of life (stand. beta = 0.23, *p* = 0.01). Conversely, a breast cancer diagnosis raises trait anxiety (stand. beta = 0.23, *p* = 0.01) but reduces both psychophysical stress (stand. beta = −0.36; *p* < 0.001) and physical quality of life (stand. beta = −0.46; *p* < 0.001). The model accounts for 39% of the variance in physical quality of life (R^2^ = 0.39) and 53% of the variance in psychophysical stress (R^2^ = 0.53).

In the third mediation model, we investigated whether psychophysical stress acts as a mediator in the relationships between trait anxiety and depression mood, as predictors, and psychophysical well-being as outcome variables. Notably, we excluded confounding variables since none of them were significantly correlated with the outcomes.

As observed from Table 8, psychophysical stress indeed mediates both the relationship between trait anxiety and psychophysical well-being and the relationship between depression mood and psychophysical well-being. Specifically, higher levels of trait anxiety and depression mood contribute to increased psychophysical stress, which, in turn, reduces the perceived psychophysical well-being. However, it is important to note that in both cases, the mediation effect is partial. Introducing the mediator does not eliminate the direct effects of the predictors on the outcome—they remain significant. The model explains 57% of the variance in psychophysical well-being (R^2^ = 0.57) while the explained variance in psychophysical stress is 49% (R^2^ = 0.49). For a visual representation and the path coefficients, you can refer to Figure 2.

The results of the mediation models applied to a sample of 193 women undergoing genetic counseling for hereditary breast cancer can be summarized as follows.

-Higher trait anxiety decreases mental quality of life both directly and through the mediation of psychophysical stress. A higher depression mood also decreases mental quality of life, entirely mediated by psychophysical stress. Education is a risk factor that slightly increases psychophysical stress and decreases mental quality of life.-Physical quality of life is reduced by higher depression mood, lower education level, and having a diagnosis of cancer. There is neither a mediation effect of psychophysical stress nor any direct or mediated effect of trait anxiety.-Psychophysical well-being is reduced by higher trait anxiety and depression mood, both directly and through the mediation of psychophysical stress. Sociodemographic and clinical confounding variables do not have any effects.

## 4. Discussion

It has been well established that women at risk of carrying a genetic mutation experience higher levels of psychological distress, particularly anxiety, compared to the general population [5,6,7,8]. However, few studies have explored the correlation between psychological distress (including anxiety and depression) and quality of life in this group [2,5,42]. Additionally, no studies have examined the relationship between psychological distress, psychophysical stress, and quality of life. This study aimed to fill this gap, and the results reveal several noteworthy associations. Psychophysical stress plays a mediating role or acts as a bridge between psychological factors like trait anxiety and depressed mood and outcomes related to mental quality of life and overall psychophysical well-being. Specifically, higher levels of trait anxiety and depressed mood lead to increased psychophysical stress, which, in turn, negatively affects perceived mental well-being and overall psychophysical health. Interestingly, psychophysical stress does not play the same mediating role when it comes to physical quality of life. Instead, depressed mood alone is a significant predictor of physical quality of life. This means that regardless of the levels of trait anxiety, it is a depressed mood that directly impacts how individuals perceive their physical quality of life. This relationship holds true even when factors like education level and breast cancer diagnosis are taken into account. These findings underscore the complex interplay between psychological factors, stress, and quality of life outcomes in the context of hereditary breast cancer screening. They highlight the importance of addressing both anxiety and depression to improve mental and overall well-being while focusing specifically on depression to enhance the physical quality of life.

To interpret these data, we can begin with established literature: the balance of these patients is marked by a complex system of individual–family interactions. Multi-generational patterns of disease manifestation can shape developmental processes and contribute to psychological distress, manifesting as anxiety and depression [43]. The study by Coyne et al. [44] has indicated that belonging to a high-risk family is perceived as more distressing than receiving genetic test results although the perception of a serious threat to one’s health is associated with significantly elevated levels of state anxiety [45].

Our findings suggest that chronic psychological distress generates a state of stress attributable to symptoms such as fatigue, tiredness, sleep disturbances, and a sense of overload. These results are consistent with the study by Gonzalez et al. [6], which found that 65.5% of patients undergoing genetic counseling express high levels of concern and one-third experience sleep problems. A portion of this population develops severe psychiatric disorders such as post-traumatic stress disorder (PTSD). In the study by Heidi et al. [31], 16.7% of women reported experiencing threshold and sub-threshold PTSD related to their personal or family history of cancer. An additional 26.2% reported a past diagnosis of PTSD. Similar findings were confirmed by Lingberg and Wellish [32], who noted that the prevalence of PTSD symptoms is comparable to that found among patients with breast cancer and other oncological diseases.

Our study results indicate that psychophysical stress, generated by emotional distress, significantly influences perceived mental well-being and overall psychophysical health. This finding builds on the work of Barbosa Oliveira et al. [5], who demonstrated that emotional distress, primarily expressed through anxiety symptoms, has a more substantial impact on overall psychophysical health and perceived mental well-being than on social relationships: increasing levels of emotional distress related to the risk of developing cancer are associated with lower perceived mental well-being. Conversely, higher levels of self-efficacy and social support are linked to better perceived mental well-being.

In our study, we found that depressed mood is the only factor that directly impacts physical quality of life. This can be explained by the fact that depression manifests through symptoms that strongly affect bodily aspects. Additionally, a depressed mood in women undergoing genetic counseling is correlated with poor motivation for care [9]. These findings are clinically significant. Emotional distress present in the pre-test phase of genetic counseling is one of the main predictors of long-term distress [21]. It can directly influence the quality of life (QoL) of patients even in the subsequent phases of genetic counseling [2,18].

The results of this study reveal a complex scenario wherein the life histories of patients, influenced by their own or their family members’ experiences with cancer, generate significant distress. This distress is characterized by emotional factors such as anxiety and depression, which contribute to a state of psychophysical stress. This stress, in turn, heightens the perception of personal vulnerability and negatively impacts the perceived quality of life. The anticipation of genetic testing results and the perceived threat to one’s health further exacerbate emotional distress, stress, and quality of life. Therefore, it is crucial to evaluate these aspects from the beginning of the genetic counseling process. Understanding these factors is essential because they can significantly influence patients’ behavioral strategies for early diagnosis and prevention.

## 5. Limitations

While this study has provided valuable insights, it is essential to acknowledge its limitations. The study likely employed a cross-sectional design, which captures data at a single point in time. As a result, it could not establish causality or infer temporal relationships. Longitudinal studies would be more informative for understanding how these factors evolve over time.

Although recruitment from two different hospitals may increase the external validity of findings, making them more applicable to diverse healthcare settings, conducting a study across only two hospitals introduces several limitations that should be considered. The findings may not apply universally; variations in patient demographics, healthcare practices, and resources across hospitals can impact generalizability; the study’s results may reflect regional characteristics or practices unique to those two hospitals (e.g., protocols, staff expertise) that may influence outcomes. In summary, while a two-hospital study provides valuable insights, its limitations underscore the need for broader research and diverse settings to enhance the robustness and applicability of findings. Furthermore, the binomial test suggests a significant difference in proportions between the two hospitals, and the small sample from the Federico II Institute (n = 22) hinders a reliable comparison of their characteristics. Smaller samples yield wider confidence intervals and less statistical power. Researchers should interpret results cautiously due to this limitation. More substantial studies with balanced samples would yield more robust conclusions.

The reliance on self-report questionnaires introduced potential biases (e.g., social desirability bias, recall bias). Objective measures (e.g., physiological markers) or other-report measures that involve collecting information from a third party could have enhanced the validity of the study.

Despite controlling for several sociodemographic and clinical factors, other unmeasured confounding variables may have influenced the observed relationships. For example, factors like socioeconomic status, social support, or a diagnosis or familiarity with other types of cancer were not explicitly addressed in this study.

Mediation models are inherently complex. While psychophysical stress mediates some relationships, other unexplored pathways may exist. Similarly, the study focused on trait anxiety and depression mood. Other psychological factors (e.g., resilience, and coping styles) may also impact the overall quality of life and well-being.

## 6. Conclusions

The study’s findings have significant implications for clinical practice.

Women undergoing hereditary screening for breast cancer may experience several psychological challenges. The process of genetic testing can evoke anxiety due to uncertainty about the results and their implications. Waiting for test results can be particularly distressing, leading to heightened anxiety levels. The possibility of carrying a hereditary mutation associated with breast cancer can lead to depressive feelings. Coping with the emotional burden of potential risk can impact overall well-being. Making decisions about risk-reducing interventions (e.g., surgery, increased surveillance) can cause stress and balancing the benefits and risks of different options can be overwhelming. Seeking psychological support during the screening process is therefore essential and healthcare providers should address these concerns and provide appropriate support to women undergoing the screening. By enhancing mental and overall well-being, patients will be better equipped to navigate the screening process.

Delivering appropriate interventions during hereditary breast cancer screening requires a comprehensive understanding of both psychological and physical factors. By integrating these aspects, healthcare providers can consider a whole person and optimize patient care and well-being. Recognizing the intricate interplay between these aspects is essential for effective screening. By adopting a holistic perspective, providers can tailor interventions to address both the biological risk (physical) and the emotional impact (psychological) of screening. Key considerations include adopting a comprehensive approach that acknowledges the interaction between psychological and physical aspects during screening. Assessing trait anxiety and depressed mood provides valuable insights into patients’ overall quality of life and well-being. It is crucial to recognize that psychophysical stress significantly impacts mental health. Implementing stress management strategies to enhance mental well-being in patients undergoing screening is recommended. For example, psychological support, counseling, and coping strategies can mitigate the impact of psychophysical stress. A recent review suggests that anxiety can be effectively treated using mindfulness-based stress reduction (MBSR). It has been shown that MBSR can improve the psychological care of breast cancer patients, both during and after treatment [46].

Although psychophysical stress affects mental quality of life and psychophysical well-being, it does not directly mediate the relationship with physical quality of life. Even after controlling for the education level and having a breast cancer diagnosis, depression mood significantly predicts physical quality of life. Therefore, clinicians should prioritize depression mood assessment and address depression symptoms during breast cancer screening.

In summary, this study has emphasized the need for integrated care, addressing psychological distress alongside physical health and recognizing the multifaceted impact of hereditary breast cancer screening on patients’ overall quality of life and well-being.

## Figures and Tables

**Figure 1 cancers-16-03613-f001:**
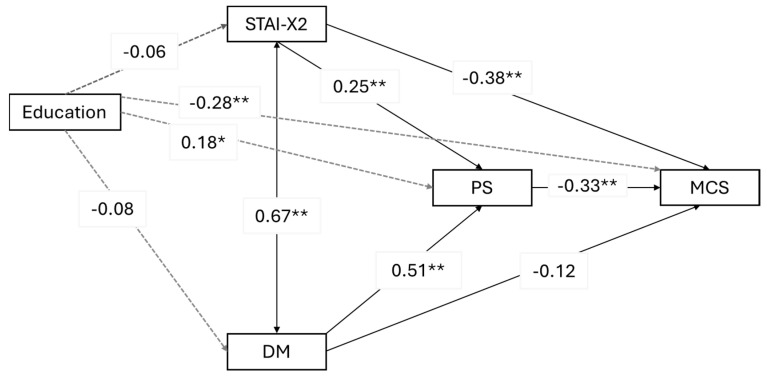
Path plot of the first mediation model. STAI-X2 = trait anxiety; PS = psychophysical stress; MCS = Mental Component Scale; DM = depression mood. * *p* < 0.05; ** *p* < 0.001. Reported estimates are standardized beta values.

**Figure 2 cancers-16-03613-f002:**
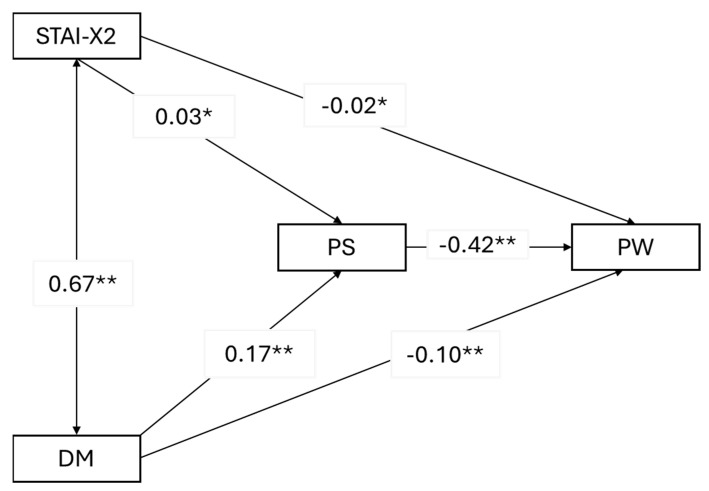
Path plot of the third mediation model. STAI-X2 = trait anxiety; PS = psychophysical stress; PW = psychophysical well-being; DM = depression mood. * *p* < 0.05; ** *p* < 0.001. Reported estimates are standardized beta values.

**Table 1 cancers-16-03613-t001:** Descriptive statistics of sociodemographic variables (N = 193).

	Minimum	Maximum	Mean	Std. Deviation
Age (years)	18	81	43.62	10.92
Relatives affected by breast cancer	0	8	2.27	1.56
	Frequency	Percent		
Education				
Primary and middle school	25	13.00		
High school	96	49.70		
University	72	37.30		
Married/cohabiting				
Yes	65	33.2		
No	128	66.3		
Having children				
Yes	67	34.7		
No	126	65.3		
Having a job				
Yes	62	32.1		
No	131	67.9		
Diagnosed with breast cancer				
Yes	95	49.2		
No	98	50.8		
Familiarity with breast cancer				
Yes	19	8.8		
No	174	90.2		

**Table 2 cancers-16-03613-t002:** Descriptive statistics of psychological variables and reliability.

Variables	Range	Mean (Std. Deviation)	Reliability ^a^
Mental quality of life (MCS)	15.08–61.71	46.57 (10.20)	0.81
Physical quality of life (PCS)	19.14–60.79	45.23 (10.22)	0.82
Psychophysical well-being (PW)	0–6	3.29 (2.04)	0.79
Trait anxiety (STAI-X2)	25–65	44.08 (8.48)	0.85
Depression mood (DM)	0–10	3.34 (2.91)	0.84
Psychophysical stress (PS)	0–7	3.05 (2.04)	0.76

^a^ Cronbach’s α.

**Table 3 cancers-16-03613-t003:** Correlations between psychological variables.

	MCS	PCS	PW	PS
STAI-X2	−0.65 *	−0.36 *	−0.59 *	0.59 *
DM	−0.59 *	−0.56 *	−0.67 *	0.67 *
PS	−0.65 *	−0.31 *	−0.70 *	-

STAI-X2 = trait anxiety; MCS = Mental Component Scale; PCS = Physical Component Scale; PW = Psychophysical well-being; DM = depression mood; PS = psychophysical stress. * *p* < 0.001.

**Table 4 cancers-16-03613-t004:** Correlations between sociodemographic and clinical characteristics and outcomes.

	MCS	PCS	PW
Age	−0.10	−0.26 **	−0.13
Education	−0.19 **	0.19 **	−0.04
Married or cohabiting	0.09	0.05	−0.02
Having children	0.13	−0.01	0.05
Having a job	−0.06	−0.02	−0.08
Diagnosed with breast cancer	−0.06	−0.39 **	−0.09
Breast cancer familiarity	0.06	0.15 *	0.02
Relatives with breast cancer	0.05	0.15 *	0.05

MCS = Mental Component Scale; PCS = Physical Component Scale; PW = psychophysical well-being. * *p* < 0.05; ** *p* < 0.001.

**Table 5 cancers-16-03613-t005:** Coefficients of multiple linear regression analysis with the dependent variable physical quality of life.

Model	Unstandardized B	Std. Error	Standardized Beta	*t*	*p*-Value
(Constant)	450.56	40.27		100.66	<0.001
Age	−0.10	0.07	−0.12	−10.47	0.14
Education	20.18	10.02	0.14	20.14	0.03
Diagnosed with breast cancer	−60.32	10.49	−0.31	−40.26	<0.001
Breast cancer familiarity	20.95	20.64	0.08	10.12	0.26
Relatives with breast cancer	0.49	0.48	0.08	10.01	0.31

**Table 6 cancers-16-03613-t006:** First mediation model: parameter estimates.

	Estimate	Std. Error	z-Value	*p*-Value	95% CI	
					Lower	Upper
Indirect effects						
STAI-X2 → PS → MCS	−0.08	0.03	−2.86	0.005	−0.14	−0.03
DM → PS → MCS	−0.17	0.04	−3.73	<0.001	−0.26	−0.08
Direct effects						
STAI-X2 → MCS	−0.38	0.08	−4.65	<0.001	−0.54	−0.22
DM → MCS	−0.12	0.08	−1.38	0.17	−0.28	0.05

CI = confidence interval; STAI-X2 = trait anxiety; PS = psychophysical stress; MCS = Mental Component Scale; DM = depression mood.

**Table 7 cancers-16-03613-t007:** Second mediation model: parameter estimates.

	Estimate	Std. Error	z-Value	*p*-Value	95% CI	
					Lower	Upper
Indirect effects						
STAI-X2 → PS → PCS	0.003	0.02	0.16	0.87	−0.04	0.04
DM → PS → PCS	0.008	0.05	0.16	0.87	−0.08	0.10
Direct effects						
STAI-X2 → PCS	0.02	0.08	0.24	0.81	−0.13	0.17
DM → PCS	−0.50	0.11	−4.71	<0.001	−0.70	−0.29

CI = confidence interval; STAI-X2 = trait anxiety; PS = psychophysical stress; PCS = Physical Component Scale; DM = depression mood.

**Table 8 cancers-16-03613-t008:** Third mediation model: parameter estimates.

	Estimate	Std. Error	z-Value	*p*-Value	95% CI	
					Lower	Upper
Indirect effects						
STAI-X2 → PS → PW	−0.01	0.004	−3.12	0.002	−0.02	−0.005
DM → PS → PW	−0.07	0.02	−4.61	<0.001	−0.10	−0.04
Direct effects						
STAI-X2 → PW	−0.02	0.008	−2.27	0.02	−0.13	0.17
DM → PW	−0.10	0.03	−3.80	<0.001	−0.70	−0.29

CI = confidence interval; STAI-X2 = trait anxiety; PS = psychophysical stress; PW = psychophysical well-being; DM = depression mood.

## Data Availability

The original data presented in the study are openly available at https://gbox.garr.it/garrbox/s/lL7mAAEhysNoK9i (accessed on 6 September 2024).

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
