# Peer review of "How Psychophysical Stress Can Mediate the Effects of Anxiety and Depression on the Overall Quality of Life and Well-Being in Women Undergoing Hereditary Breast Cancer Screening"

_cancers, 2024, doi:10.3390/cancers16213613_

Round 1

Reviewer 1 Report

Comments and Suggestions for Authors

Dear authors,

I have reviewed your manuscript, and overall, the introduction seems appropriate. However, the novelty of this study compared to previous research is not entirely clear. It would be beneficial to clarify the unique contribution of your study.

Additionally, it could be helpful to mention other variables that are commonly studied in cancer survivors, such as stress and anxiety (e.g., 10.3390/jcm12226968). Highlighting these aspects might provide further context for your research.

Regarding the methods section, I would appreciate more information on the steps taken to control for confounding variables. Furthermore, I did not fully understand the rationale behind using the "The Cognitive Behavior Assessment – Hospital form" in your study.

In the results section, a concluding paragraph emphasizing the key findings would be valuable to strengthen the overall presentation of your results.

Lastly, the discussion appears adequate.

Thank you for considering these suggestions.

Comments on the Quality of English Language

Minor editing of English language required.

Author Response

1 I have reviewed your manuscript, and overall, the introduction seems appropriate. However, the novelty of this study compared to previous research is not entirely clear. It would be beneficial to clarify the unique contribution of your study.

Answer: thank you for this important suggestion. We have clarified this point at the end of the Introduction (before the Objectives paragraph).

2 Additionally, it could be helpful to mention other variables that are commonly studied in cancer survivors, such as stress and anxiety (e.g., 10.3390/jcm12226968). Highlighting these aspects might provide further context for your research.

Answer: as you suggested, we have mentioned other studies that addressed some of the variables we used in cancer survivors (line 65-68).

3 Regarding the methods section, I would appreciate more information on the steps taken to control for confounding variables.

Answer: we explained more in depth the approach to control for the confounding variables in both the Analysis section and the Results section (in red).

4 Furthermore, I did not fully understand the rationale behind using the "The Cognitive Behavior Assessment – Hospital form" in your study.

Answer - We used the CBA-H because it provides a brief measure, already validated in Italy, of three of the main variables in our study. Depression mood that we considered along with anxiety an expression of distress and a predictor; Psychophysical wellbeing that we used together with quality of life to have a more comprehensive outcome; and Psychophysical stress that we used as the mediator in the relationship between distress and quality of life and wellbeing. Moreover, the CBA-H has been recently used in the context of genetic counseling for hereditary breast cancer. We have added this explanation to the description of the tool in the Measures section.

5 In the results section, a concluding paragraph emphasizing the key findings would be valuable to strengthen the overall presentation of your results.

Answer: thank you very much. We have added this concluding paragraph to clarify and strengthen the presentation ofour  results.

6 Lastly, the discussion appears adequate.

Answer: thank you for this positive evaluation.

7 Minor editing of English language required.

Answer: we have revised the English language throughout the manuscript.

Reviewer 2 Report

Comments and Suggestions for Authors

Quality of life and well-being are insufficiently highlighted in the introductory part. A better conceptualization from a psycho-oncological perspective would be desirable.

However, the theoretical part is sufficiently synthetic but adjustable.

The procedure, measurements and data analysis are presented succinctly and efficiently.

Generally "participants" are not presented in the results section. I suggest a combination of the sections working procedure and participants.

The results section is presented scientifically and succinctly.

The discussions and conclusions are well written.

Author Response

1 Quality of life and well-being are insufficiently highlighted in the introductory part. A better conceptualization from a psycho-oncological perspective would be desirable.

Answer: we have added a more detailed description of both quality of life and psychophysical well-being (lines 78-85 and 111-123).

2 However, the theoretical part is sufficiently synthetic but adjustable.

Answer: Thank you, we adjusted the theoretical part. We also expanded the description of the experience of stress that we used as mediator in our model.

3 The procedure, measurements and data analysis are presented succinctly and efficiently.

Answer: Thank you very much for this positive comment.

4 Generally "participants" are not presented in the results section. I suggest a combination of the sections working procedure and participants.

Answer: Most articles published in Cancers report data of participants in the Results section, under “Participants” (e.g., https://doi.org/10.3390/cancers16203477; https://doi.org/10.3390/cancers16203462 etc.). We tried to combine more information in the Procedure section, but we maintained the data of the sample in the Participants section under the Results in line with the other articles in the same journal.

5 The results section is presented scientifically and succinctly. The discussions and conclusions are well written.

Answer: thank you for this evaluation.

Round 2

Reviewer 1 Report

Comments and Suggestions for Authors

Dear authors,

For me, the paper is suitable for publication.

Best regards.